# Integrating competency-based assessment into training with peer counsellors providing support at scale to children, adolescents and young adults living with HIV in Zimbabwe: An implementation pilot case study

**Carol Wogrin** [1]*, **Sarah Bernays**[2], **Shelter Dhliwayo**[1], **Eliza Gwenzi**[1], **Debra Machando**[3], **Charity Mandimika**[1], **Shepard Munyoro** [1], **Tanyaradzwa Napei**[1], **Getrude Ncube**[4], **Billiart Tapesana**[1], **James Underhill** [5], **Nicola Willis** [1]

1 Zvandiri, Harare, Zimbabwe, 2 School of Public Health, University of Sydney, Sydney, Australia, 3 Independent Consultant, Harare, Zimbabwe, 4 Ministry of Health and Child Care, Harare, Zimbabwe, 5 Independent Consultant, Brighton, England, United Kingdom

* carol@zvandiri.org

## Abstract

Zvandiri is a differentiated psychosocial support program delivering peer-counselling to children, adolescents and young adults living with HIV (CAYALHIV), functioning at a national scale in Zimbabwe and in 14 other African countries. Stigma and mental health issues are significant drivers of suboptimal health outcomes in CAYALHIV, particularly in low-resource settings where the professional workforce is inadequate to meet the volume of need. Task shifting to peer counsellors has shown promise in improving care yet developing robust mechanisms to ensure consistent quality has received limited attention to date. To promote quality services delivered by peer counsellors (ages 18 – 24 years), Zvandiri sought to pragmatically address the feasibility and effectiveness of integrating the World Health Organization's EQUIP (Ensuring Quality in Psychological Support) competency-based assessment into a standard peer-counsellor training. Sixteen CATS, ages 19 – 22 years (M8:F8), already working as Zvandiri peer-counsellors, were identified by their supervisors as needing skills strengthening and selected to participate in the standard five-day CATS training. Additional days were added for baseline and endline competency assessments. This pilot integrating EQUIP's competency-based approach into the CATS training effectively identified gaps in counselling skills at baseline, enabling trainers to target these competencies throughout the training. The endline assessment demonstrated that all 16 participants developed stronger counselling skills. A follow-up open-text self-assessment was conducted seven months post-training with both participants and supervisors. There was consistency in the reporting of increased confidence and improved client interactions, with changes attributed to the enhanced counselling skills learnt in the training. Based on these findings, Zvandiri will integrate EQUIP assessments into the initial training provided to peer-counsellors when they first join Zvandiri and within the CATS' regular group supervision, in a targeted way to support the identification of further

**Data availability statement:** Relevant data is available via the Figshare database: https://figshare.com/articles/dataset/Integrating_competency-based_assessment_into_training_with_peer_counsellors_providing_support_at_scale_to_children_adolescents_and_young_adults_living_with_HIV_in_Zimbabwe_an_implementation_pilot_case_study/28382522

**Funding:** The author(s) received no specific funding for this work.

**Competing interests:** The authors have declared that no competing interests exist.

skills strengthening required to maintain quality delivery of counselling services at scale by peer-counsellors.

## Author summary

Zvandiri is a differentiated psychosocial support program delivering peer-counselling to children, adolescents and young adults living with HIV (CAYALHIV) to improve HIV and mental health outcomes, functioning at a national scale in Zimbabwe and in 14 other African countries. To promote and maintain quality services delivered at scale by peer counsellors (ages 18 – 24 years), Zvandiri sought to pragmatically address the feasibility and effectiveness of integrating the World Health Organization's EQUIP (Ensuring Quality in Psychological Support) competency-based assessment into a standard peer-counsellor training program. Baseline and endline assessments using EQUIP were conducted. The pilot of the EQUIP's competency-based approach effectively identified gaps in counselling skills on baseline assessment, enabling trainers to target these competencies throughout the training. The endline assessment demonstrated that all 16 participants developed stronger foundational counselling skills. Based on these findings, Zvandiri will integrate EQUIP assessments i) into the initial training provided to peer-counsellors when they first join Zvandiri and ii) within the CATS' regular group supervision, in a targeted way to support the identification of further skills strengthening required to maintain the quality delivery of counselling services at scale by peer-counsellors.

## Introduction

Positive health outcomes among children, adolescents and young adults living with HIV (CAYALHIV) continue to lag compared to outcomes for adults. This is particularly true for young people living in low-resource settings [1,2,3]. Problems related to stigma and discrimination, risk-taking sexual behaviours, early pregnancies, gender-based violence, substance use, difficulties with adherence to antiretroviral therapy and retention in care are among the plethora of challenges faced [4,5]. Additionally, high rates of depression and other mental health problems are well documented [6] and are both cause and consequence of the problems which characterise the adolescence phase for many CAYALHIV [4,6,7].

However, there is a significant gap between the level of need and available mental health services due to the low number of mental health professionals in low and middle-income countries (LMIC) and substantial ongoing resource-constraint. This results in the under-identification of mental health problems and inadequate services for those in need [4,5]. Recognition of this gap has prompted efforts to shift community level mental health service delivery from mental health professionals to trained lay counsellors. There is a growing body of evidence that this task shifting is a feasible and effective way to address the high level of need [8,9].

While the potential of this approach is promising, to date there is little known about how to best support ongoing quality control in the delivery of care at scale, and almost none for youth-led care. One of the major questions for effective task-shifting, is how best to train and supervise primary care workers, considering the issue of how to reliably and validly assess adequate competency to safely and effectively provide care [10]. In LMICs, laypersons often come from more deprived socioeconomic backgrounds and have minimal formal education. Therefore, in-depth training, supervision and support from mental health professionals is required for task-shifting to be successful and to ensure high quality care [11]. Yet, for non-specialists

there are no mechanisms in place to systematically monitor whether providers achieve minimally adequate competency to effectively and safely deliver interventions [12].

Recognizing that a training on counselling skills that focuses only on the delivery of knowledge with no assessment of the acquisition of knowledge is inadequate in and of itself to ensure quality services, WHO and UNICEF developed EQUIP (Ensuring Quality in Psychological Support) to improve the assessment of the effectiveness of task shifting first-line mental health care to lay counsellors [13]. EQUIP provides tools for assessment and supportive resources for organizations and trainers to utilize. It is designed to strengthen the quality of trainings and guide trainers and supervisors using a competency-based approach to ensure that participants can demonstrate the skills needed to deliver high quality counselling services [14]. EQUIP tools identify a series of competencies, and for each competency, four levels of skills: unhelpful or potentially harmful counselling behaviours (level 1), foundation skills (levels 2 and 3) and advanced skills (level 4) in counselling. The tools enable trainers and supervisors to assess and provide feedback on the level of skill demonstrated across the specific counselling competencies.

Current research conducted on EQUIP with adult counsellors has shown that compared to standard programs, participants in EQUIP based trainings were more likely to reduce harmful behaviours and strengthen helping skills [13] and be effective in promoting the development of counselling skills [15,16].

Zvandiri is an effective, differentiated service delivery peer-support model supporting the Ministry of Health and Child Care (MoHCC) in the treatment, care and support of CAYALHIV and is being implemented at a national scale in Zimbabwe and within 13 other African countries. Currently, across Zimbabwe, there are approximately 1,100 CATS who support 60,000 CAYPLHIV. Detailed descriptions of the Zvandiri intervention have been published previously [8,17], but the Zvandiri program is outlined below. Central to Zvandiri's model is task-shifting between mental health specialists and the peer counsellors, which has been shown to be an effective approach to addressing the gaps in mental health care for CAYALHIV [18,19]. In addition to expanding the workforce, this shift in frontline care to peer counsellors has advantages in terms of what peer support offers, since young clients are frequently more likely to open up to people close to their age and with similar life experiences (Table 1).

The peer counselling provided by the Zvandiri CATS to CAYALHIV plays a critical health system role to support optimal health outcomes. The challenge to Zvandiri is ensuring quality and safe delivery of services at scale. To work effectively, CATS need strong counselling skills to identify and discuss the myriad of psychological and social factors that contribute to poor HIV outcomes in children, adolescents and young adults. Zvandiri needs a training approach that provides adequate, targeted capacity building, systematic support and oversight for the peer counsellors that can be implemented feasibly and cost-effectively. EQUIP's competency-based assessment approach was piloted within Zvandiri to assess whether its use could improve the quality of the CATS' counselling skills and to support the maintenance of quality over time. EQUIP has the potential to be a valuable tool to support the ongoing capacity strengthening of CATS delivering the Zvandiri program, and if effective with this cohort of peer counsellors, provide the mechanism to identify areas needing strengthening in programming on the organizational level, and in supervision on the individual level.

To inform whether to implement EQUIP in an integrated and systematic way, we undertook a pragmatic pre and post-test analysis of the impact of the EQUIP assessment tool with a selected group of Zvandiri's peer counsellors (CATS). In this paper we report on the pragmatic assessment of the feasibility of delivery and its effectiveness in supporting sustained and high-quality peer-delivered counselling at scale across the program.

**Table 1. Zvandiri program methods.**

| Community Adolescent Treatment Supporters | CATS are ages 18 – 24 years. They can become CATS between the ages of 18 – 22 years. |
|---|---|
| Recruitment | Recruited jointly by Ministry of Health and Child Care (MOHCC) and Zvandiri within the facilities where they receive their HIV care. |
| Eligibility criteria for selection to be a CATS | • Young people, 18-22 years old<br>• Living with HIV on ART, virally suppressed<br>• Committed to supporting peers<br>• Understands and is ready for Zvandiri Peer Counsellors roles and responsibilities<br>• Clinically and psychosocially well<br>• Not in school<br>• Can read and write<br>• Approval from primary caregiver |
| Selection | • Interviewed jointly by the facility HCW and Zvandiri District team, using:<br>- Zvandiri selection interview tool<br>- Zvandiri health check tool<br>- Mental health screening tool |
| Roles and responsibilities | • Support the health and well-being of children, adolescents and young people living with HIV through differentiated case management<br>• Services provided in homes, facilities, and through digital health, weekly/ monthly, according to their clinical and psychosocial needs, supported by CATS Supervisor.<br>• CATS are assigned their clients by the facility HCWs.<br>• CATS provide ongoing care until either they graduate from being a CATS, or their client turns age 25 and transitions to adult care. |
| Training | • CATS receive an initial 5-day standardized residential training based on WHO Global Standards for Quality Health Care Services for Adolescents [20].<br>• The training is comprised of 15 interactive modules focused on HIV literacy, care and treatment, counselling skills, SRHR and other adolescent health related topics.<br>• Following the 5-day training, CATS are placed in facilities where they receive further capacity building from the MoHCC HCWs and the Zvandiri District Teams (ZDTs).<br>• Trainings and subsequent work is supported by a wide range of tools, many available on Zvandiri's website (www.zvandiri.org) |
| Supervision | • CATS meet weekly with the HCWs in their facilities who oversee their caseload to discuss physical and mental treatment needs of individual clients.<br>• CATS meet monthly with the ZDT for mentorship, skill building and case discussions.<br>• All CATS within a district come together quarterly, for capacity building. |

## Methods

### Ethics statement

An ethics waiver (MRCZ/E/339) was provided by the Medical Research Council Zimbabwe (MRCZ) for the anonymised program data related to the training, including the numerical assessment of their pre and post-test competency levels, to be published so that program implementation evidence can contribute to the limited evidence-base. The waiver was granted on the grounds that this pilot was aimed at improving and evaluating standard Zvandiri programming.

The Zvandiri Mentors identified 16 CATS, ages 19 – 22 (M8:F8), from Harare who had been CATS for approximately a year and were understood to be in need of additional training support to strengthen their counselling skills and confidence to participate in a CATS training incorporating the competency-based assessments. They were invited to participate and all agreed. There would have been no ramifications had they chosen not to. Written individual

consent was not sought as this constituted part of program training. The follow-up check-in was part of standard program practice to support ongoing appraisal of impact.

## Process of adapting EQUIP training for target population

In order to tailor the EQUIP assessment tool to be best placed to meet the needs of child, adolescent and young adult beneficiaries (up to age 24), an adaptation process was undertaken in collaboration between Zvandiri and WHO to identify the most salient age-appropriate competencies for the Zvandiri target population and program. Through virtual meetings, discussions were held about the target age ranges for the EQUIP competency tools, which include the ENACT and WeACT tools. ENACT is for use with adults, ages 19 and older, and WeACT is for use with children and adolescents up to age 18. While CATS are ages 18 – 24 years, Zvandiri recipients of care (RoC), for whom they provide counselling, are ages birth – 24 years, with many RoCs being adolescents and young adults. Following the discussion and with WHO approval, a hybrid of the two tools was developed, comprised of 14 competencies. Ten competencies were drawn from ENACT. Several ENACT competencies were excluded due to being aimed at a higher developmental level than many Zvandiri RoCs, and have a focus on clinical care, including diagnosis, treatment and care plan, that is not applicable to the peer counselling provided by CATS. The 10 competencies included from ENACT are:

1. Non-verbal communication & active listening

2. Verbal communication skills

3. Explanation & promotion of confidentiality

4. Rapport building & self-disclosure

5. Exploration & normalisation of feelings

6. Exploration of client's & social support network's explanation for problem (causal & explanatory models)

7. Appropriate involvement of family members & other close persons

8. Promotion of realistic hope for change

9. Incorporation of coping mechanisms & prior solutions

10. Psychoeducation & use of local terminology

Four competencies were included from WeACT, due to their developmentally appropriate wording and their relevance to the work of the CATS.

1. Empathy, warmth & genuineness

2. Safe identification of child abuse, exploitation, neglect, violence, & self-harm

3. Giving feedback to the child

4. Acknowledges & promotes child's agency in the session

## Integrating the EQUIP assessment into training

The training was implemented in June 2023. In addition to Zvandiri staff who facilitated the training and conducted the assessments, each of whom had counselling training either as a social worker, nurse counsellor, or clinical psychologist, the training was attended by the WHO Zimbabwe Country Office Mental Health National Professional Officer and the

Strategy and Innovations Officer from Pamumvuri, an organization supporting the WHO Country Office in integrating EQUIP into mhGAP trainings. Their role was to provide guidance to Zvandiri on the integration of EQUIP. The standard 5-day training was conducted over nine days. Three days were added to conduct the pre-assessment (categorised as our pre-test), midline and endline assessments (the latter forming our post-test result), in line with the EQUIP training approach. The standard CATS training is residential, and the days are long. For this pilot, CATS travelled to and from the head office, requiring shorter days. An additional day was added to the schedule to accommodate for the shorter days. Day one of the CATS training was run as usual since it orients participants to the role of CATS and the experience of living with HIV, rather than counselling skills.

### Pre-training assessment (pre-test)

On day 2 of the training, CATS were divided into four groups to conduct pre-training assessments to establish the baseline skill level of each participant. In each group of four, one person role played a CATS, one roleplayed a client, one was an observer, and one videotaped the role-play. A training facilitator was present in each group to assess the CATS across the 14 competencies. The participant role-playing the client was provided a brief description of the client they were to play, and the client's main problem. The CATS conducted a 10-minute role play of a first counselling session with their beneficiary, following which the facilitator rated the competencies. The participants then changed roles and a role-play was run with a different CATS and client so that each participant was videotaped conducting a roleplay in which they were the peer counsellor, and their competencies were rated.

Feedback was held until all four participants had been assessed, so that each demonstrated their baseline skill level without the advantage of learning from feedback given to others. After the four participants had completed their roleplays they were given feedback one at a time, highlighting points from the across the competencies that they did well and identifying several specific areas for them to work on. For example, if they had failed to describe limits to confidentiality, or in attempts to reassure a client they minimized a client's feelings, this was explained to them. If they had demonstrated any potentially harmful behaviours, these were the points chosen to highlight. Feedback to each ended by again emphasising points that had been done well. These competency ratings constitute the pre-test (baseline) assessment.

### Midline and endline competency assessments (post-test)

Following the baseline roleplays, the next day and a half of the standard CATS training was conducted. These were days that included a focus on counselling skills. The afternoon following the baseline assessments the Zvandiri team and the WHO and Pamumvuri representatives viewed and rated two of the baseline videos. Discussion was had on each person's rating of the 14 competencies to establish inter-rater agreement and ensure that the four team members rated the CATS consistently. At the end of week one, midline assessments were conducted using the same approach of dividing the CATS into groups of 4 for roleplays, videotaping and assessments. In week two, days 3 – 5 of the CATS 5-day training were conducted, and then a final day of role plays, videotaping and assessments was done to assess their endline skills. Based on the behaviours observed, the level of each of the 14 competencies was rated. The level achieved for each of the competencies at endline served as the post-test results. These were compared with the levels they had achieved on baseline assessments, and were shared with the CATS, emphasising each competency where the CATS had demonstrated higher level than they had on baseline assessment.

Because Zvandiri used a hybrid of ENACT (adult tool) and WeACT (child and adolescent tool), we were not able to use WHO's EQUIP online platform. Zvandiri's monitoring and evaluation team developed Zvandiri's database for the EQUIP assessments using KoboTools at the recommendation of WHO.

### Follow-up assessment

Following the training, CATS continued with Zvandiri's standard programming including receiving supervision, which is provided weekly by the health care workers (HCWs) in their facilities, where they discuss the cases that the CATS are supporting, and monthly by Zvandiri District Teams (ZDTs). Once each quarter, all CATS in a district come together for CATS Coordination meetings with the Zvandiri District Teams for mentorship.

The 16 CATS who participated in the training were contacted seven months post-training via a WhatsApp group by the training facilitator. The rationale for this timeframe was that any transient benefits of the training would have faded, and there would have been ample time for them to have integrated new learning into their practice. The training participants were asked to reflect on four questions about their experience with the training, and inbox their answers individually to the facilitator: 1) did the training change anything in the way you do your peer counselling? 2) If yes, what do you do differently now? 3) What difference does this make with your beneficiaries? 4) What most stands out when you think back on the training? A simple content analysis was conducted on their anonymised written responses to assess maintenance and sustained effectiveness. The data were categorised into a priori determined categories relating to the competencies that were the focus of the training. The ZDT was contacted as well, for feedback on whether they saw any lasting changes in the participating CATS.

### Findings

This training focused on 16 CATS who had been identified by their supervisors as requiring support to strengthen their counselling skills and confidence. At baseline, most of the behaviours demonstrated by the CATS were level 2 basic skills, with the majority of the CATS also demonstrating some unhelpful or potentially harmful behaviours. Of the 16 CATS, 12 demonstrated 1 – 3 unhelpful behaviours each (out of a possible 45 across the 14 competencies), with 22 unhelpful behaviours in total observed in the group. None of the CATS demonstrated advanced level skills on any of the competencies at baseline.

At midline assessment, following the training days focused on counselling skills, only one CATS demonstrated level 1 behaviours (potentially harmful counselling behaviours) on two competencies. At endline, none of the CATS showed any level 1 behaviours. Across all competencies there was a shift from level 2 behaviours to level 3 (foundational levels) between baseline and midline, with the exception of giving feedback to the child, where the shift from level 2 to level 3 wasn't seen until endline assessments. Advanced level competencies markedly increased during the training, with none demonstrated at baseline and 12 of the CATS demonstrating advanced level skills at endline (Fig 1).

The two competencies that were the most challenging for the CATS, as determined by the potentially harmful behaviours at baseline, were exploration and normalization of feelings (from ENACT) and empathy, warmth and genuineness (from WeACT) (Figs 2 and 3). The summary of these competencies across the assessment points are shown below. The specific behaviours that were challenging were related to minimizing or ignoring clients' feelings and expressions of distress. At times when a client got upset, the CATS responded by quickly trying to assure the client in an effort to help the client feel better, whether by offering unrealistic hope, e.g. 'you'll be fine', 'you won't die from HIV' (attempting to sooth a client upset

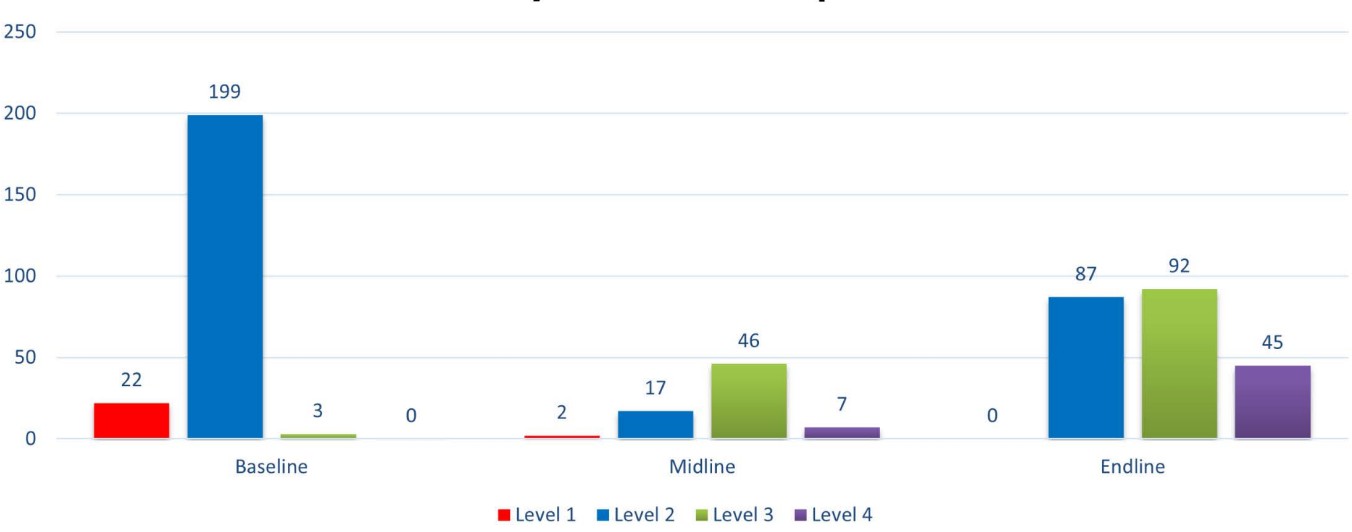

**Fig 1.  Summary of the 14 Competencies** .

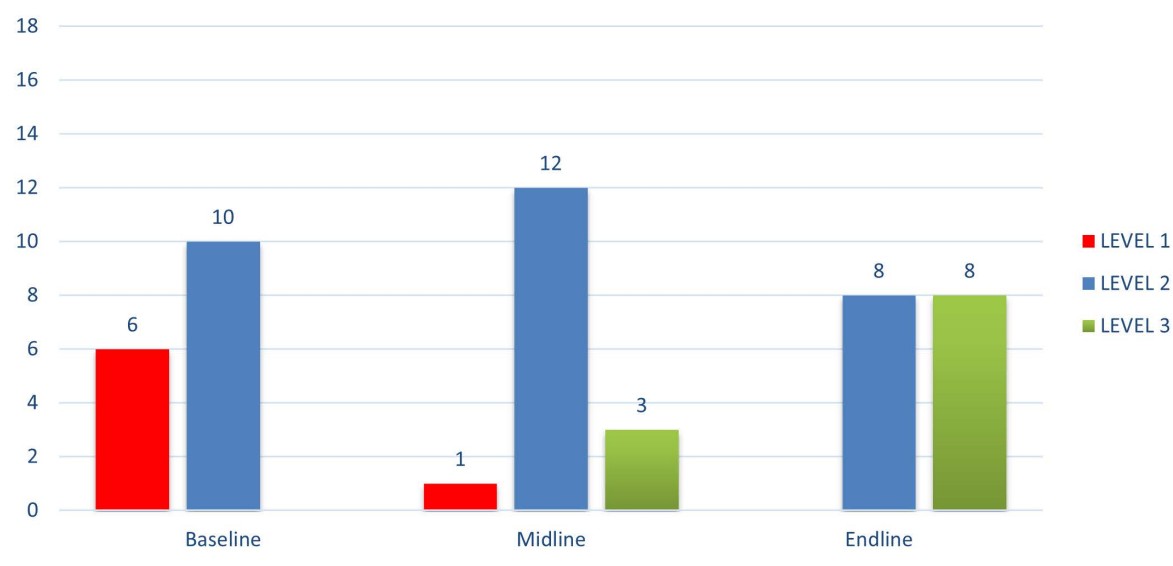

**Fig 2.  Exploration & Normalizaion of Feelings** .

that she might die from HIV the way her mother had), or minimizing feelings, 'don't worry, everything will be ok' (to a client upset having learned about their positive HIV status and another upset after a girlfriend ended the relationship upon learning of the client's positive HIV status). Incorporating a focus on these competencies through roleplays and feedback, where behaviours such as dismissing a client's emotions, or not responding to them were pointed out and new approaches were practiced, helped the CATS understand their counselling behaviours and develop the skills to handle them differently.

## 6. EMPATHY,WARMTH & GENUINENESS(from WeACT)

**Fig 3. Empathy, Warmth & Genuineness .**

Two other competencies, verbal communication skills and explanation of confidentiality were of particular note in terms of the results achieved by taking a competency-based approach to strengthening foundational counselling skills. Compared to baseline, where CATS demonstrated levels 1 and 2 behaviours, by the end of the training, they were demonstrating more level 3 and level 4 behaviours, routinely using phrases such as, "Can you tell me more about that?" or "If I understand you correctly, you're saying…" (Fig 4).

Further, while CATS were familiar with the principal of confidentially and explaining this to clients, they were less likely to have good language for explaining the limits of confidentiality. Being young peer-counsellors, they work under the principal of shared confidentiality. They are part of the team in the health facilities where they practice, and work under the supervision of the facility HCWs. Cases, concerns, and treatment plans are routinely discussed. Informing clients of the need to share information with supervisors, and the limits of confidentiality when there is risk of serious harm proved challenging. By the end of the training incorporating EQUIP, CATS showed marked improvement in their skill level with this concept, moving from predominantly level 2 skills, to mostly levels 3 and 4 at endline (Fig 5).

*Self-confidence.* Seven months after the training, the CATS were contacted by the lead training facilitator via WhatsApp and asked to reflect in writing on their experience of the training in response to the four questions outlined in the methods. They all responded within 24 hours (with the exception of one who had problems with phone connectivity). All spoke favourably about what they felt they had gained in the training. While the responses were not anonymous, the level of specificity in their responses indicated that they were not just giving platitudes. A number of the CATS stated that it boosted their confidence because they had a better understanding of counselling skills, with one CATS stating, "*I learned new techniques for active listening and providing support, and I feel more confident in my ability to help others.*

*Affective connection.* They described an increased understanding of some of the competencies, and their ability to apply what they learned:

## 2. VERBAL COMMUNICATION SKILLS

**Fig 4. Verbal Communication Skills .**

## 3. EXPLANATION & PROMOTION OF CONFIDENTIALITY

**Fig 5. Explanation & Promotion of Confidentiality .**

*"I would say that I approach peer counselling with more empathy and compassion since the training... I'm more aware of my body language and tone, and I try to be fully present and focused on the person I'm helping. I've also started using more open-ended questions to encourage people to share more about their experiences."*

Specifically addressing the challenge of tolerating a client's difficult emotion that was demonstrated on baseline assessment, one CATS described the following:

*Before the training i was that type of counsellor who didn't have the courage enough to talk to the client/adolescent, i was limited by what to talk about, what to say to make my client/*

*adolescent feel better after the session, i was overwhelmed by my own fear, But after the Equip Training……I came along with a client who was having mental and social issues in which when i was talking to her she started crying and i gave her time to cry and show her emotions, I listened to everything she said without jumping into her story.*

Difficulty tolerating a client's pain was also seen in baseline assessments when CATS were quick to reassure, which at times resulted in encouraging hope that wasn't realistic. CATS recognized that they had changed their approach in this way as well. "*Also, we as CATS we do not promise anything to our beneficiaries such like, 'all your problems will be gone trust me, don't worry,' but we give hope to our beneficiaries.*"

**Strengthening relationships.** Finally, they recognized that the changes in these skills resulted in changes in the ways that their clients relate to them, "*They seem to feel more heard and understood, and they're more likely to share their experiences and feelings with me. They also seem more willing to try new coping strategies and make positive changes in their lives, I think they're benefiting from my new approach to peer counselling.*"

The ZDTs' appraisals aligned with the CATS' self-reports, stating that significant improvement was seen in all of them, with five of the 16 standing out over time. These five had gone from demonstrating skills that were weaker than those of their peers, to exceeding the skills of their peers in terms of identification of recipients of care who were in need of increased support, offering care in keeping with Zvandiri guidelines, and referring those with higher levels of distress for further management.

## Discussion

The Zvandiri program has been shown to deliver effective peer-based counselling to improve CAYALHIV's HIV outcomes and reported mental health [17,19,21,22]. Yet there are concerns about how to ensure fidelity and quality control at scale, including about how to provide sufficient ongoing support and capacity building within the constraints of restricted donor funding.

In this implementation pilot, using EQUIP assessment tools to strengthen the Zvandiri program training, we found that it supported targeted training and supervision, guiding decisions about how to support peer-counsellors to advance their counselling skill development. Our use of EQUIP provides valuable learning about the potential positive impact of competency-based training to effectively support skill development in a way that is deliverable, assessable, and scalable.

Using the targeted approach with the CATS not only helped the individual peer counsellors develop their skills, but also helped trainers get a greater and more nuanced understanding of areas where more explanation and practice was needed. The specific feedback during the training, beginning with their areas of strength and then highlighting areas to focus on for further development, helped make the somewhat abstract concepts that are the basis for counselling skills more concrete, and therefore more attainable for the CATS. As they practiced, conducting roleplays and receiving feedback, their understanding of desirable behaviours grew, with this their confidence grew as well.

A focus on verbal and non-verbal active listening skills is standard in counselling curricula, but putting these concepts into practice takes time, effort, and attention. While CATS had been introduced to these concepts in their initial training, the implementation of EQUIP provided them with specific, targeted feedback. It also gave them the opportunity to observe their peers demonstrating counselling skills in roleplays and learn from that feedback as well, strengthening their understanding of these concepts and applying the concepts to their own behaviours.

Peer counsellors demonstrated their ability to develop advanced counselling skills using this method. The use of the competency-based assessments helped trainers identify the specific challenges faced by peer counsellors, such as the natural tendency to want to "fix" a problem for a client, or quickly comfort them to make them feel better, adapt the training accordingly, and then focus more heavily on the skills of empathy and normalization of feelings. The articulation of specific behaviours across the levels helped make the underlying concepts actionable for the CATS and their understanding of what they needed to do and say improved. As they gained skill with active listening and appropriately encouraging expression of feelings, they also got more comfortable with supporting the emotions expressed, demonstrating that these skills can be strengthened through the tailored training that EQUIP provides.

Assessing the competencies also helped identify areas in which the CATS needed help with specific language to communicate necessary information. This was seen particularly in their difficulty with explaining the limits to confidentiality. The CATS had trouble finding language that described sharing information with others, fearing that telling a client they would share information from their discussions would prevent a client from opening up. Identifying this through the competency assessment and roleplay observation, trainers were able to focus on this skill with the CATS, suggest language and have the CATS practice in roleplays. By the end of the training, these young peer counsellors were demonstrating solid basic counselling skills and some advanced skills.

Following the training, a number of the CATS were able to reflect on what they were doing differently, including using more open-ended questions, tolerating pain and fears in their clients, allowing for the expression of emotion, tolerating silences and letting the client go at their own pace. They saw the effect this had on their relationships with their clients and the impact on their work. EQUIP's competency-based approach provided them with a framework for observing themselves as well as their clients.

Based on the positive outcome of this implementation pilot the EQUIP training will be incorporated into future training through two feasible pathways, which can be achieved at minimal cost. The first will be within the initial 5-day training that all new CATS receive when they begin as peer counsellors. Although delivered across a number of sessions, in total this will add an extra day to the training, as focus on specific competencies and roleplays will be integrated into the current modules and activities that comprise the training. An additional day will provide time to complete assessments at the end of the training. Zvandiri is primarily donor funded, and CATS trainings, supervision and CATS coordination meetings and are already funded as an integral part of implementation of the program in the districts. The only additional cost will be this addition of a day in the CATS initial training. The assessments, which identify which competencies an individual needs further strengthening, will be shared with the district teams where the CATS will be working. The competency assessments will also be incorporated into current CATS' regular group supervision, which will require little additional investment and cost.

The pre and post assessment of the CATS' competencies in the delivery of general counselling, based on observed behaviours, allows for comparison over time and between individuals. The follow-up self-reported data was corroborated by observations of their supervisors, indicating that the training has an ongoing effectiveness. The impact of this quality counselling on beneficiaries was not the subject of enquiry but has been the subject of previous research highlighting its effectiveness [19]. The small number of participants assessed in this initial pilot of the integration of EQUIP into training limits the generalisability of our findings. However, as the selection criterion was to identify those who would benefit from further training to enhance their competence and not based on any other criteria, i.e. their demographic profiles otherwise reflect the patterns within the wider CATS' cohort, the training is likely to have benefit to the

cohort of CATS more broadly. As EQUIP was implemented and integrated into the training being delivered in an ongoing program, we were able to assess feasibility. There are multiple opportunities to continue to assess the value of this training approach through further implementation. The opportunity to include this learning within the published literature, due to the provision of an ethics waiver, highlights the value of program implementation learning.

## Conclusion

Integrating EQUIP competency assessment into the training of older adolescent and young adult peer counsellors is both feasible and effective to strengthen their counselling skills. The competency-based approach helped target specific skills needing emphasis during the training and identify areas requiring further skill development, which could then be integrated into regular supervision. Zvandiri is now in the process of capacity building with staff across Zimbabwe with the aim of integrating EQUIP into CATS trainings as standard practice and into ongoing supervision. We will continue to monitor and evaluate the use of EQUIP as it is integrated into programming to further inform its use to promote counselling competency among Zvandiri peer counsellors. This approach holds significant potential for supporting task shifting and maintaining the safety and quality of services delivered at scale, which is a vital component to redressing widespread unmet mental health needs in resource-constrained settings.

## Acknowledgments

We thank the CATS for their participation and efforts, Zvandiri staff who provided the training, the World Health Organization and the Ministry of Health and Child Care, Zimbabwe, for their support in developing and implementing this program initiative.

## Author contributions

**Conceptualization:** Carol Wogrin, Debra Machando, Billiart Tapesana, James Underhill, Nicola Willis.

**Data curation:** Carol Wogrin, Tanyaradzwa Napei.

**Formal analysis:** Carol Wogrin, Tanyaradzwa Napei.

**Investigation:** Carol Wogrin, Shelter Dhliwayo, Eliza Gwenzi, Charity Mandimika, Shepard Munyoro.

**Methodology:** Carol Wogrin, Billiart Tapesana.

**Project administration:** Carol Wogrin.

**Supervision:** Carol Wogrin.

**Writing – original draft:** Carol Wogrin, Sarah Bernays, Debra Machando, James Underhill, Nicola Willis.

**Writing – review & editing:** Carol Wogrin, Sarah Bernays, Shelter Dhliwayo, Eliza Gwenzi, Debra Machando, Charity Mandimika, Shepard Munyoro, Tanyaradzwa Napei, Getrude Ncube, Billiart Tapesana, Nicola Willis.

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
