## [Decision Letter · Decision Letter 0]

6 Nov 2024

PMEN-D-24-00292

Integrating competency-based assessment into training with peer counsellors providing support at scale to children, adolescents and young adults living with HIV in Zimbabwe: a case study

PLOS Mental Health

Dear Dr. Wogrin,

Thank you for submitting your manuscript to PLOS Mental Health. After careful consideration, we feel that it has merit but does not fully meet PLOS Mental Health’s publication criteria as it currently stands. Therefore, we invite you to submit a revised version of the manuscript that addresses the points raised during the review process.

We look forward to receiving your revised manuscript.

Kind regards,

Abigail Mae Hatcher, PhD

Academic Editor

PLOS Mental Health

Journal Requirements:

**Please only choose the relevant sentences from below**

1. Please clarify all sources of funding (financial or material support) for your study. List the grants (with grant number) or organizations (with url) that supported your study, including funding received from your institution. 

2. State the initials, alongside each funding source, of each author to receive each grant.

3. State what role the funders took in the study. If the funders had no role in your study, please state: “The funders had no role in study design, data collection and analysis, decision to publish, or preparation of the manuscript.”

4. If any authors received a salary from any of your funders, please state which authors and which funders.

2. In the online submission form, you indicated that "Data is stored on Zvandiri’s Figshare. Upon reasonable request, data can be

accessed through contacting Vivian Chitiyo". 

3. Uploaded as supplementary information.

3. Please insert an Ethics Statement at the beginning of your Methods section, under a subheading 'Ethics Statement'. It must include:

1) The name(s) of the Institutional Review Board(s) or Ethics Committee(s)

2) The approval number(s), or a statement that approval was granted by the named board(s) 

3) (for human participants/donors) - A statement that formal consent was obtained (must state whether verbal/written) OR the reason consent was not obtained (e.g. anonymity). NOTE: If child participants, the statement must declare that formal consent was obtained from the parent/guardian.

4. Please provide an Author Summary. This should appear in your manuscript between the Abstract (if applicable) and the Introduction, and should be 150–200 words long. The aim should be to make your findings accessible to a wide audience that includes both scientists and non-scientists. Sample summaries can be found on our website under Submission Guidelines:

https://journals.plos.org/mentalhealth/s/submission-guidelines#loc-parts-of-a-submission

5. Please provide separate figure files in .tif or .eps format.

https://journals.plos.org/mentalhealth/s/figures 

https://journals.plos.org/mentalhealth/s/figures#loc-file-requirements 

Additional Editor Comments (if provided):

Kindly revise the paper in direct response to reviewer concerns. Since Reviewer 2 was not specific about their suggested changes, I might suggest you consider stating the research question more clearly (at the end of the Introduction). In Methods, please do elaborate on the type of data collected, how this was analyzed, who was involved in analysis. Lastly, in Limitations within Discussion, you might note potential shortcomings of this choice in methods and how future research could overcome these particular constraints.

Reviewers' comments:

Reviewer's Responses to Questions

**Comments to the Author**

1. Does this manuscript meet PLOS Mental Health’s publication criteria ? Is the manuscript technically sound, and do the data support the conclusions? The manuscript must describe methodologically and ethically rigorous research with conclusions that are appropriately drawn based on the data presented.

Reviewer #1: Yes

Reviewer #2: No

2. Has the statistical analysis been performed appropriately and rigorously?

Reviewer #1: N/A

Reviewer #2: I don't know

3. Have the authors made all data underlying the findings in their manuscript fully available (please refer to the Data Availability Statement at the start of the manuscript PDF file)?

Reviewer #1: No

Reviewer #2: No

4. Is the manuscript presented in an intelligible fashion and written in standard English?

Reviewer #1: Yes

Reviewer #2: Yes

5. Review Comments to the Author

Reviewer #1: I would like to applaud the Zvandiri group for their really interesting and important work. I found the article very engaging, interesting, and adds to the literature on effective task-shifting for addressing this vulnerable population of AYALWH in Zim.

I have a few comments that might strengthen the paper, mostly in adding context to improve understanding the program. Though there are references cited, it will be helpful to include a bit of detail to strengthen the reader’s understanding if the program:

More about the program in the methods:

1) It is clear that CATS are 18-24 years of age. It is less clear at what age they become a CATS? How is one chosen to be a CATS? What happens when they exit the program…. when they turn 25? How are CATS compensated? How the # of CATS fluctuates.

2) Do CATS use a training manual or is there a manual of content they are supposed to deliver? There is a mention of how to ensure fidelity…..fidelity to what exactly? To counseling skills?

3) How do clients find CATS? How are CATS paired with clients and what is their goal/objective? Are they paired for a short time, a long time, how many sessions, etc? Do they meet at the clinic, at the client’s household, elsewhere?

4) What exactly does weekly supervision by HCW and monthly by ZDT entail?

5) Is ethics approval needed to report on training-competency results?

Results:

6) The data on integrating EQUIP into training is very compelling, but this was for 16 youth that were identified as needing further training. Can you elaborate on their demographics: how old, what gender, how long had they been working as a CATS? Could this be in comparison to the other CATS that are working, but were not chosen for this particular training?.....i.e. were these 16 younger, a certain gender, less experienced?

7) These 16 CATS were selected based on their need for ….did the supervisors who selected them comment on their skill, improved or similar post-training (there is data from CATS, but not from the supervisors who suggested they attend the training). That perspective could be helpful.

Discussion

8) There is mention that EQUIP will be incorporated into future Zvandiri trainings. How expensive is this addition? How often do trainings occur given CATS presumably graduate and need to be replaced? How many CATS need to be trained to keep the program quality at scale? Who pays? There is mention of being challenged by constraints in donor funding. Is there room for the MOH in Zimbabwe to support the program? Or what is the path of sustainability?

9) Limitations are not discussed. Suggest there are a few limitations in that the N is small, focused on those needing additional support—do all or a few when you consider cost and scale up? if supervisors did not provide input I would consider that a limitation of results, similarly, the perspective of the client (did they feel helped) is missing.

Reviewer #2: The idea behind this paper is needed and important, unfortunately in it's current state it is unclear whether the manuscript speaks to a specific research question and methods as well as the purpose, scope, and results of the study are not clearly described. For instance, qualitative methods and analysis are not fully described in the methods; how coding was done; or how consent was obtained. The qualitative findings help support quantitative findings given the small sample size, but as currently written the research questions being answered by the study are not. Although not suitable for publication with major revisions at this point, we are hopeful the authors might be able to better restructure the paper to highlight the gap in research being filled, the research question, and how it was answered and why.

6. PLOS authors have the option to publish the peer review history of their article (what does this mean? ). If published, this will include your full peer review and any attached files.

**Do you want your identity to be public for this peer review?** For information about this choice, including consent withdrawal, please see our Privacy Policy .

Reviewer #1: No

Reviewer #2: No

---

## [Editor Report · Decision Letter 1]

6 Mar 2025

Integrating competency-based assessment into training with peer counsellors providing support at scale to children, adolescents and young adults living with HIV in Zimbabwe: an implementation pilot case study

PMEN-D-24-00292R1

Dear Dr Wogrin,

We are pleased to inform you that your manuscript 'Integrating competency-based assessment into training with peer counsellors providing support at scale to children, adolescents and young adults living with HIV in Zimbabwe: an implementation pilot case study' has been provisionally accepted for publication in PLOS Mental Health.

Best regards,

Abigail Mae Hatcher, PhD

Academic Editor

PLOS Mental Health

This is ready for publication.